# Wireless, Portable Fiber Bragg Grating Interrogation System Employing Optical Edge Filter

**DOI:** 10.3390/s19143222

**Published:** 2019-07-22

**Authors:** Ken Ogawa, Shouhei Koyama, Yuuki Haseda, Keiichi Fujita, Hiroaki Ishizawa, Keisaku Fujimoto

**Affiliations:** 1Graduate School of Science and Technology, Shinshu University, 3-15-1 Tokida, Ueda, Nagano 386-8567, Japan; 2Faculty of Textile Science and Technology, Shinshu University, 3-15-1 Tokida, Ueda, Nagano 386-8567, Japan; 3Research & Development Sec. Development Center, Naganokeiki Co., Ltd., 2150 Ikuta, Ueda, Nagano 386-0411, Japan; 4Institute for Fiber Engineering, Shinshu University, 3-15-1 Tokida, Ueda, Nagano 386-8567, Japan; 5Department of Clinical Laboratory Sciences, School of Health Sciences, Shinshu University, 3-1-1 Asahi, Matsumoto, Nagano 390-8621, Japan

**Keywords:** fiber bragg grating, vital sign monitoring, optical edge filter, pulse wave, plethysmograph

## Abstract

A small-size, high-precision fiber Bragg grating interrogator was developed for continuous plethysmograph monitoring. The interrogator employs optical edge filters, which were integrated with a broad-band light source and photodetector to demodulate the Bragg wavelength shift. An amplifier circuit was designed to effectively amplify the plethysmograph signal, obtained as a small vibration of optical power on the large offset. The standard deviation of the measured Bragg wavelength was about 0.1 pm. The developed edge filter module and amplifier circuit were encased with a single-board computer and communicated with a laptop computer via Wi-Fi. As a result, the plethysmograph was clearly obtained remotely, indicating the possibility of continuous vital sign measurement.

## 1. Introduction

The necessity of continuous vital sign monitoring has been increasing in many aging societies. As sensing equipment is downsized, continuous vital sign monitoring such as electrocardiograph (ECG) [1], phonocardiograph (PCG) [2] and photo-plethysmograph (PPG) [3,4] have become readily available. In such applications, users wear these monitors, so the whole system needs to be small, lightweight and portable.

The fiber Bragg grating (FBG) sensor is a strong candidate for such vital sign monitoring applications. The FBG is a diffraction grating inscribed in the core of an optical fiber, which reflects a specific wavelength, called the Bragg wavelength, of incident light [5]. The Bragg wavelength is changed with the strain and temperature induced on the FBG [5,6,7]. Because they have electromagnetic immunity and high sensitivity, FBG sensors have been used in a variety of applications such as structural health monitoring [8,9,10], vibration sensing under high electromagnetic noise [11,12] and as hydrophones [13,14]. Further, FBG sensors are much thinner than conventional electrical sensors, with almost the same diameter as a hair, and can be woven into textiles [15,16]. Considering these advantages, several studies have investigated the use of FBG sensors in vital sign monitoring [17,18,19,20,21,22]. For example, ballistocardiograph (BCG), seismocardiograph (SCG) and respiration are measured using Magnetic Resonance Imaging (MRI) [23,24,25], in which the sensors are exposed to strong electromagnetic fields. Our unit also previously worked on analyzing the waveform of vital signs, with the intention of measuring blood pressure [26,27,28,29], blood glucose levels [30] and multiple vital signs [31,32]. The goal of these studies was to achieve continuous monitoring in the future.

A lot of methods have been researched and developed to measure FBG sensors [6,7]. These methods can be briefly classified into three types: (1) wavelength-swept filters, (2) interferometers and (3) passive filters. Firstly, the (1) wavelength-swept-type interrogators employ a wavelength-swept light source or measurement window [33,34]. This method has the advantage of being able to measure many FBGs simultaneously because it has an intrinsic wavelength division multiplexing (WDM) scheme. The (2) interferometer-type interrogators employ a kind of interferometer, such as q Mach–Zehnder [8,35] or Michelson interferometer [36]. This has the great potential for measuring Bragg wavelength with extremely high precision [8,36]. These two methods have already been used in commercially available FBG interrogators. However, these methods have difficulty being downsized. Because the main application of the FBG sensors so far are for industrial or seismic sensing [8,9,10,11,12], this difficulty has not been a problem. However, when it comes to a continuous vital sign monitoring system, the size and weight of the measurement system is crucial. 

In this study, we employed a (3) passive filter [37,38,39] to develop a portable FBG interrogation system. This system employs an edged filter as the wavelength measuring element, and is small and easy to use, with high precision. The limitation of this type is the limited range of Bragg wavelengths; however, this disadvantage is not a problem for plethymogram, because the signal amplitude is totally within their range. A signal amplification circuit was also developed, which effectively amplifies the plethysmograph signal, obtained as small vibrations of optical power on a large offset. The circuit also has the output port of the offset, which contains the information of the absolute value of the FBG wavelength. With this output, the conversion coefficients of the reflect/transmit ratio to wavelength are obtained. The developed system can interrogate two FBG sensors simultaneously with a 1-kHz sampling rate and high repeatability (approximately 0.2 pm). With this interrogator, a plethysmograph of a brachial artery can be clearly obtained. The measured data are wirelessly sent to a laptop PC. A small, commercially available portable battery powers the interrogator, hence the whole system becomes portable.

## 2. Edge Filter-Based FBG Interrogator

### 2.1. Principle of Edge Filter-Based FBG Interrogation

The FBG is the periodical refractive index change inscribed in the optical fiber core. Figure 1 shows the schematic diagram of the FBG, and its effect on the incident light. The incident light propagates through the optical fiber core, and the FBG reflects only the specific wavelength of incident light determined by Equation (1),
(1)$$\lambda_{B}=2n_{\mathrm{eff}}d,$$
where $\lambda_{B}$ is the Bragg wavelength, $n_{\mathrm{eff}}$ is the effective refractive index and d is the grating interval. Physical stress such as strain, tension, pressure and temperature cause $n_{\mathrm{eff}}$ and d to shift, which are measured as a shift in $\lambda_{B}$.

To demodulate the Bragg wavelength, a lot of methods [6,7,8,33,34,35] have been suggested. In this study, an optical edge filter was employed to achieve a simple measurement setup [37]. The optical edge filter is a kind of dielectric thin film, which is normally used as a WDM filter in a variety of applications such as telecommunications and sensor multiplexing. In such applications, a stopband and passband are used. On the other hand, the slope of the transmission/reflection ratio between the stopband and the passband is used for sensing [37]. Figure 2b shows a typical wavelength dependence of the transmission/reflection ratio between the stopband and the passband of the edge filter. In this study, the slope of the edge filter is used to observe the Bragg wavelength change (in Figure 2b, indicated with a dashed square). Figure 2a shows the schematic diagram of typical experiment setups of the FBG interrogation. Because the transmission/reflection ratio of the edge filter is nearly a linear function of the wavelength, $\lambda_{B}$ is expressed as follows:
(2)$$\lambda_{B}=\alpha D+\beta ,$$
where D=(T−R)/(T+R) is the normalized differential signal [37] in which *T* and *R* denote the transmission and reflective light power of the edge filter, respectively, which is measured on the PDs. α and β are the constants determined on each edge filter. The specific values are calculated in Section 3.

### 2.2. Characteristics of the Developed FBG Interrogator

Firstly, throughout this study, the FBG (SU-CW-90-2-15-10-U-A-2-2R, Shinkodensen co.ltd., Kagawa, Japan) was used as a specimen, which was inscribed in a PANDA-type polarization-maintaining (PM) fused silica optical fiber. A single-mode fiber is more cost effective; however, it could not be used here because the edge filters have polarization-dependent loss, and the SLD (Super Luminescent Diode) emits polarized light. For continuous vital sign monitoring, we developed the edge filter module and the amplifier circuit (Figure 3, schematic diagram Figure 4). This module contains a super luminescent diode (SLD, the spectrum is shown in Figure 1), five photo-detectors (PD, KPDE030-SW, KYOTO SEMICONDUCTOR Co., Ltd., Kyoto City, Japan), a WDM filter, a half mirror (a 1:1 beam splitter which has polarization dependent loss less than 3%), a physical contact type standard connector (SC/PC) pigtail and two edge filters. Here, the WDM filter is used for just dividing two FBGs to measure simultaneously. The incident light emitted from the SLD is transmitted through the pigtail and reflected by the FBG, then the reflected light counter-propagates and, finally, is observed by the PDs. The PDs convert the reflected light to the electrical current:
(3)y=ax,
where y is the electrical current, a is conversion coefficient and x is incident light power. Each of these currents enter the amplifier circuit, shown in Figure 4b. The current-voltage converter converts this current signal to the voltage:
(4)z=by,
where z is output voltage of the current-voltage converter, and b is conversion coefficient (Figure 4b). In our previous study, it was revealed that the plethysmograph signal we intend to measure was as much as one-thousand times smaller than the offset power [29]. Hence, if the signal is simply amplified, the amplified voltage would easily exceed the range of the analog-to-digital converter (ADC) and the intended signal would not be sufficiently amplified. To overcome this difficulty, a low-pass filter (LPF) and differential amplifier is employed, hence the dynamic signal is largely amplified separately.
(5)$$z_{\mathrm{off}}=\frac{1}{\tau }\int_{t-\tau }^{t}z\left(t^{\prime }\right){dt}^{\prime },$$
(6)$$z_{\mathrm{dyn}}=c\left(z-z_{\mathrm{off}}\right),$$
where $z_{\mathrm{off}}$ is the offset power of z, $z_{\mathrm{dyn}}$ is the amplified dynamic signal, τ is the time constant of the low-pass filter and c is the amplification factor of the differential amplifier, which is set to 100. The offset power $z_{\mathrm{off}}$ is also output (Figure 4) and used to evaluate the conversion formula between the normalized differential signal to the wavelength (Figure 5). The data acquisition device (USB6210, National Instruments Corporation, Austin, TX, USA) and LabVIEW are used to measure and analyze the signal. The OSA is used to measure the Bragg wavelength corresponding to each normalized differential value obtained with the edge filter module and developed amplifier circuit. To evaluate the coefficient of Equation (2), a wavelength-tunable FBG filter is used (Figure 6). Figure 6 shows the relationship between the wavelength and the normalized differential signal. The dashed lines indicate the approximations $\lambda_{B}=1.3321\;D+1542.9567$ and $\lambda_{B}=-1.6497\;D+1560.5514$ for 1543 and 1561 nm-centered edge filters, respectively.

To obtain the amplified dynamic signal as in the unit of wavelength, the following equations are used. In the application of the plethysmogram, which is the main objective of this interrogator, repeatability (standard deviation) of the measured wavelength is crucial. The standard deviation of measured FBG wavelengths, which are in static circumstance, are 0.13 and 0.08 pm for 1543 and 1561 nm-centered FBGs, respectively. In a previous study, it was confirmed that the wavelength change caused by the plethysmogram on the radial artery was several picometers to several tens of picometers [26], so we assume that the interrogator developed in this study is applicable to the plethysmogram measurement.

## 3. Application to Vital Sign Monitoring

### 3.1. Principle and Setup

In this section, we tried to measure the plethysmograph on a brachiral artery. To analyze the data, a Butterworth-type bandpass filter was employed to reduce the noise and compare the waveforms. To achieve a remote and portable measurement system, a single-board computer (Zynq-7010, Xilinx, San Jose, CA, USA) and smaller AD board (High-Precision AD/DA Board, Waveshare Electronic, Shenzhen, China) were used to measure the output voltage of the edge filter module. These boards were encased in a case made by a 3D printer (Figure 7). The size of the case was 74×57×90 mm, and the weight was 230 g, which was small enough to be portable. The power consumption was about 0.8 W. This system only measured the $z_{T,\mathrm{dyn}}$ and $z_{R,\mathrm{dyn}}$ shown in Equation (6), from which the wavelength shift is approximated:
(7)$$\Delta \lambda =\frac{\alpha \{(z_{T,\mathrm{dyn}}-z_{R,\mathrm{dyn}})-\left(z_{T,\mathrm{dyn}}^{0}-z_{R,\mathrm{dyn}}^{0}\right)\}}{c(z_{T,\mathrm{off}}-z_{R,\mathrm{off}})},$$
where $z_{T,\mathrm{dyn}}^{0}$ and $z_{R,\mathrm{dyn}}^{0}$ are the initial values of $z_{T,\mathrm{dyn}}$ and $z_{R,\mathrm{dyn}}$, respectively; $\alpha ,\;z_{T,\mathrm{off}}$ and $z_{R,\mathrm{off}}$ are determined in the experiment in Figure 5c,d. Here, $\alpha =1.3321,\;z_{T,\mathrm{off}}=0.08966$ and $z_{R,\mathrm{off}}=0.07881$ for the 1543-nm-centered FBG, and $\alpha =-1.6497,\;z_{T,\mathrm{off}}=0.13802$ and $z_{R,\mathrm{off}}=0.23218$. c = 100 is an amplification coefficient determined in the amplifier circuit. The data were sent to a laptop PC via Wi-Fi (a USB dongle was employed on a single-board computer (GW-USNano2, PLANEX COMMUNICATIONS INC., Tokyo, Japan)) and analyzed in LabVIEW. The FBG interrogator developed in this study was used for vital sign monitoring; Figure 8 shows the measurement setup of this experiment. The 1543 and 1561 nm-centered FBG sensors were taped with surgical tape on the brachial artery of the subject (inset of Figure 8). Another 1550 nm-centered FBG inscribed on single-mode fiber was also used as reference, and measured with a commercially available FBG interrogator (PF20, Naganokeiki Co., Ltd., Nagano, Japan).

### 3.2. Results and Discussion

Figure 9 shows a plethysmograph obtained on the brachial artery with three FBGs, two for the developed interrogator with wavelengths centered to 1543 (FBG1) and 1561 (FBG2) nm, and another one centered to 1550 nm (reference). The protocol of this study was approved by the Ethics Committee of Shinshu University (Project identification code: No. 3202, Verification clinical trial with wearable vital sign measurement system). The subject was a 27-year-old male without any health problems. The sampling rate was 1 kHz for both interrogators. The pulse wave peaks were clearly obtained at the same time, and their amplitudes were about 5 pm and consistent with each other.

To obtain more details of the waveform, the data was processed with the Butterworth bandpass filter (order 2, low-cutoff frequency 0.5 Hz, high-cutoff frequency 5 Hz). The filtered plethysmogram is shown in Figure 10. Qualitatively, the FBG1 and FBG2 show similar waveforms, which indicate two local maximums next to the pulse wave peaks. The reference FBG also shows a similar tendency; however, the repeatability is worse because of the higher standard deviation, which is apparently shown in Figure 9, calculated as 0.1 pm.

## 4. Conclusions

A small, high-precision FBG interrogator was developed and evaluated. The interrogator employed an edge filter module with two wavelength channels, 1543 and 1561 nm. By dividing the low-frequency and high-frequency signals and amplifying them independently, the standard deviation of the measured wavelength was suppressed to 0.13 pm for the 1543-nm-centered FBG and 0.08 pm for the 1561-nm-centered one. The developed interrogator was applied to plethysmograph measurement on the brachial artery. As a result, the peaks of the heartbeat were clearly obtained, and their calculated amplitude of wavelength change was consistent with the reference FBG sensor. To transmit the data through Wi-Fi, we used a commercially available single-board computer, then the size of the whole interrogator was enlarged. However, the edge filter module itself has large potential to be reduced in size. The developed system indicates the viability of continuous, wearable vital sign monitoring in the future.

## Figures and Tables

**Figure 1 sensors-19-03222-f001:**
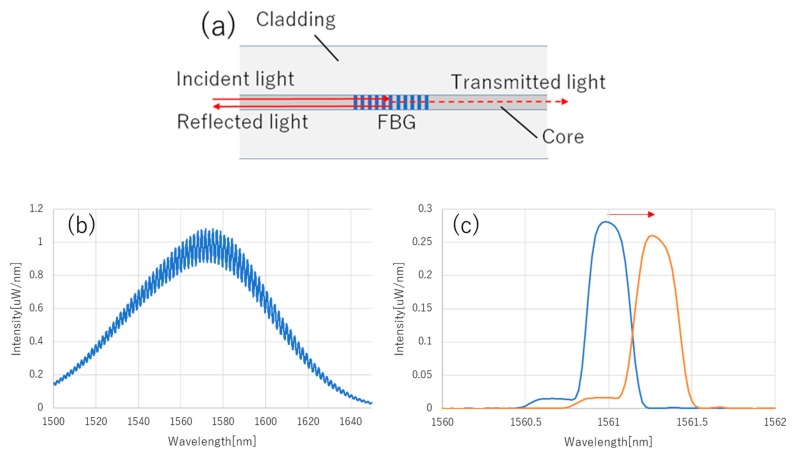
(**a**) The schematic diagram of the FBG sensor. The incident light propagates through the optical fiber core, which is surrounded by cladding. FBGs have a periodic refractive index change inscribed on the optical fiber core. An example of the incident light (**b**), and the real reflected spectrum of FBG and its change against tension (**c**) are shown in the graph. These spectrums are obtained with the optical spectrum analyzer (OSA) (AQ6370D, Yokogawa Electric Co., Tokyo, Japan) with a resolution setting of 0.1 nm.

**Figure 2 sensors-19-03222-f002:**
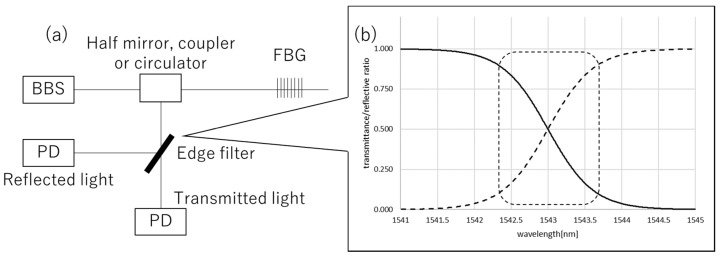
(**a**) Typical setup of the edge filter-based FBG interrogation. The light emitted from a broad band light source (BBS) enters the FBG and only the light of the Bragg wavelength is reflected. The reflected light enters the edge filter and is divided into the reflected and transmitted light, whose ratio depends on the wavelength, as in (**b**). These lights are observed by photo-detectors (PDs). (**b**) Conceptual diagram of the wavelength dependence of the transmission/reflection ratio of the edge filter. The dashed square indicates the slope used for FBG demodulation. The solid and dashed lines indicate the transmittance and reflectance ratio, respectively. The FBG wavelength is set in the dashed square range, in which the wavelength dependence could approximate a linear function.

**Figure 3 sensors-19-03222-f003:**
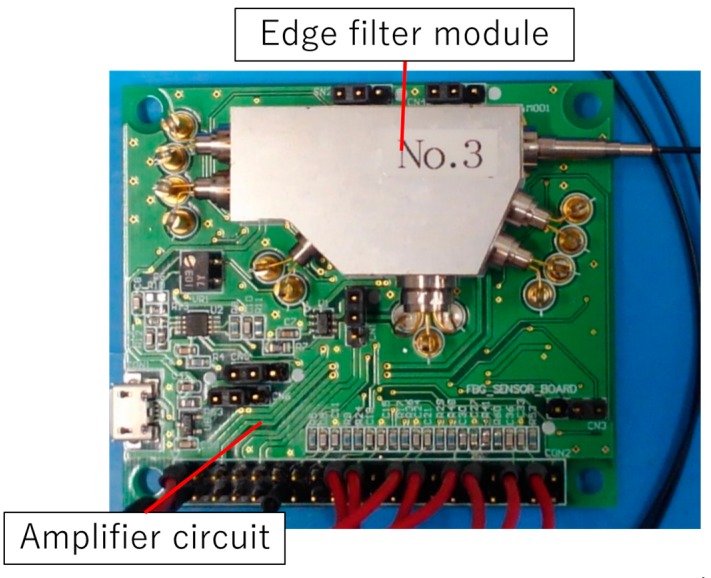
The edge filter module and amplifier circuit. The module contains an SLD, five PDs, a WDM, a half mirror, a SC/PC-pigtail and two edge filters.

**Figure 4 sensors-19-03222-f004:**
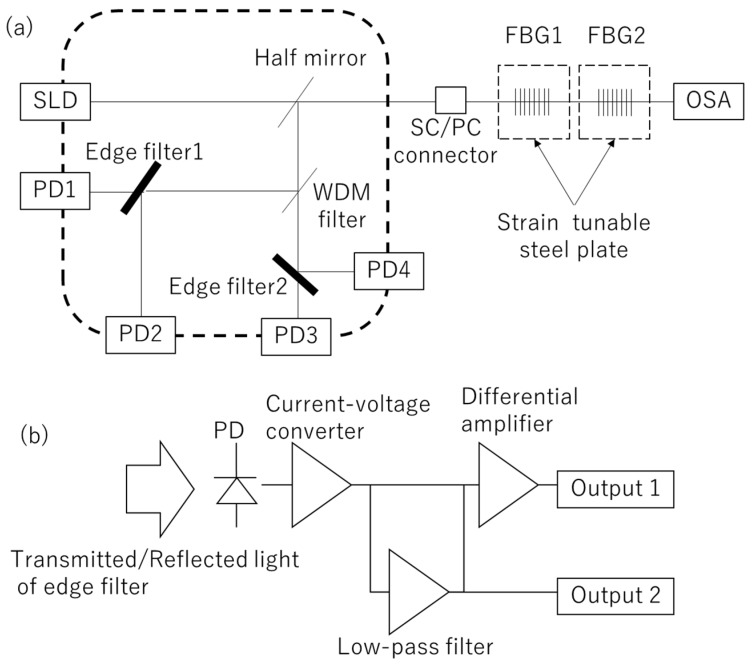
Schematic diagram of the edge filter module and amplifier circuit. (**a**) The schematic diagram of the developed edge filter module. In addition to Figure 2a, this module contains a WDM filter to measure two FBGs with different center wavelengths. Corresponding to each FBG, two edge filters are used. (**b**) The schematic diagram of the amplifier circuit. Each PD converts the light into an electrical current, then the current-voltage converter converts this into voltage. This voltage is divided into two paths: the first path enters the low-pass filter (LPF) to output the slower signal (output 2). The other is amplified by the differential amplifier with the output voltage of the low-pass filter to obtain a largely amplified faster signal (output 1).

**Figure 5 sensors-19-03222-f005:**
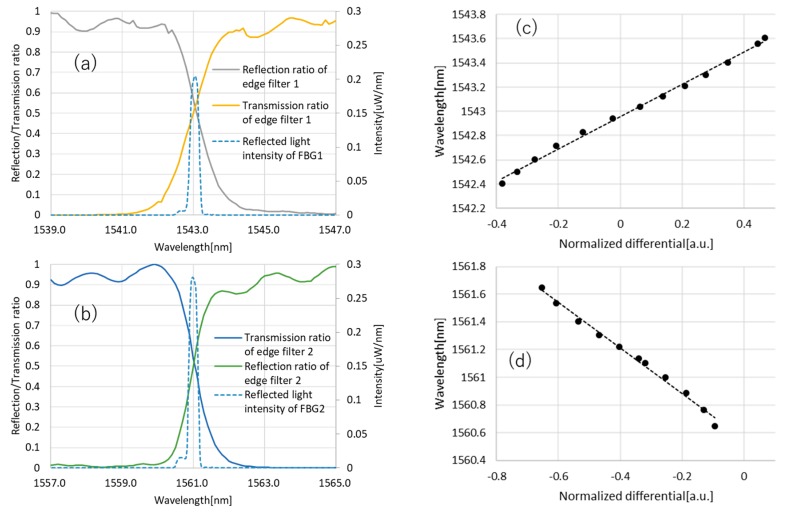
(**a**,**b**) The spectrums of the (**a**) 1543 and (**b**) 1561 nm-centered edge filters and FBGs. (**c**,**d**) The results of the Bragg wavelength vs normalized differential signal of the (**c**) 1543 and (**d**) 1561 nm-centered edge filters. Dashed lines indicate the linear approximation of the data. The formulas are $\lambda_{B}=1.3321\;D+1542.9567$ and $\lambda_{B}=-1.6497\;D+1560.5514$ for the 1543 and 1561 nm-centered edge filters, respectively.

**Figure 6 sensors-19-03222-f006:**
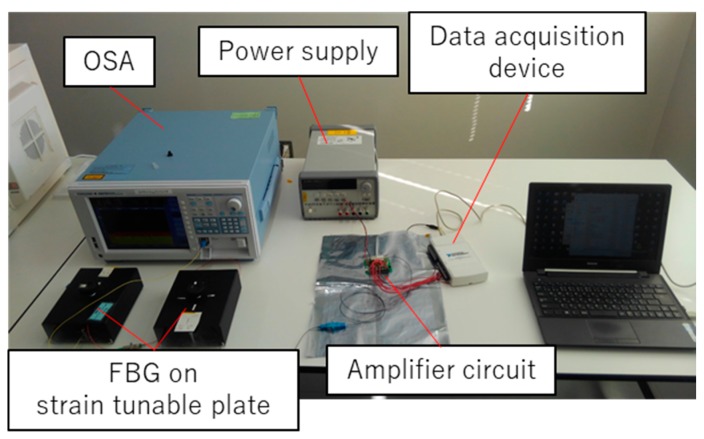
The measurement setup of the experiment to obtain the coefficient of the normalized differential to wavelength conversion formula, with 1543- and 1561-nm-centered FBGs glued separately on a steel plate, with the strain is controlled manually. The transmitted light of the FBGs is measured using an OSA, on which the Bragg wavelengths are calculated.

**Figure 7 sensors-19-03222-f007:**
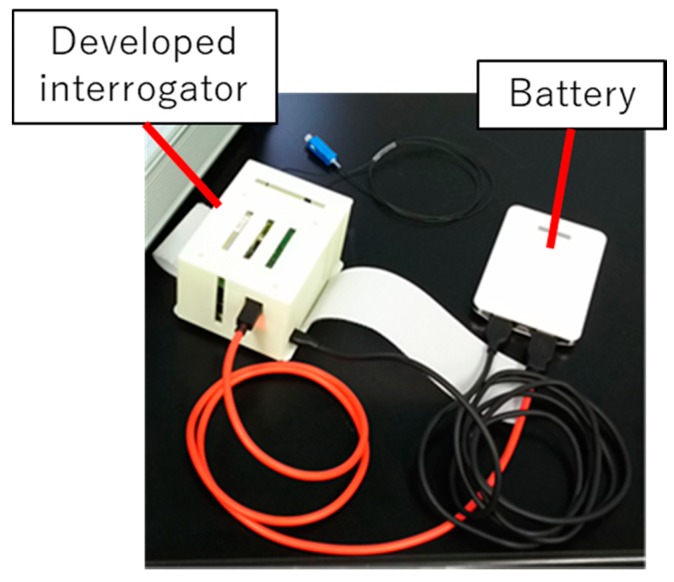
The casing of the interrogator (left). The edge filter module, single-board computer and smaller AD board are encased. A portable battery (right) supplies the power.

**Figure 8 sensors-19-03222-f008:**
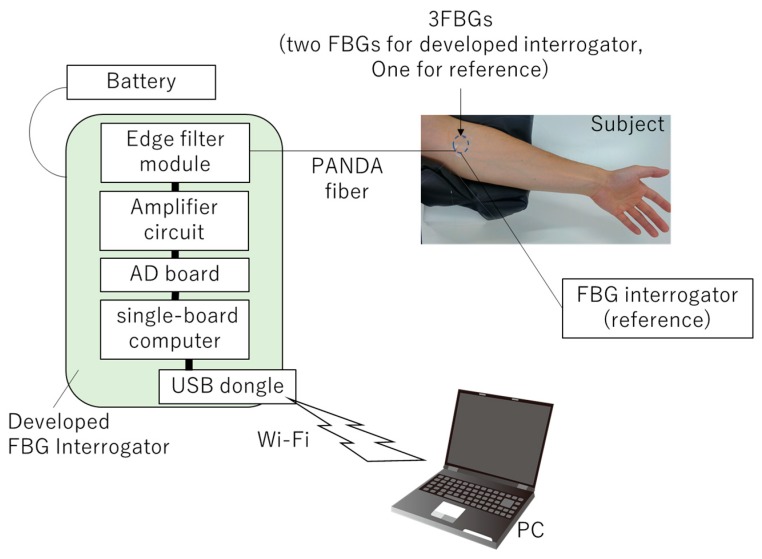
The settings of the plethysmograph measurement. The FBG sensor is taped near the brachial artery. Measured data are saved on a PC, which is connected to the interrogator via Wi-Fi.

**Figure 9 sensors-19-03222-f009:**
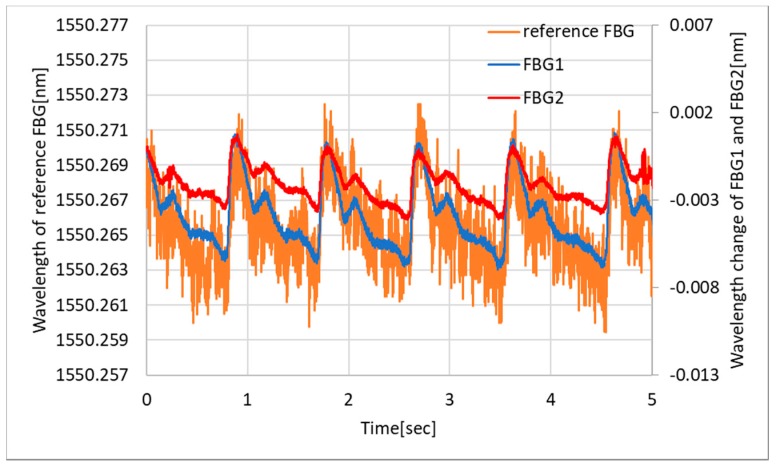
Plethysmogram obtained with the developed FBG interrogator and commercially available interrogator.

**Figure 10 sensors-19-03222-f010:**
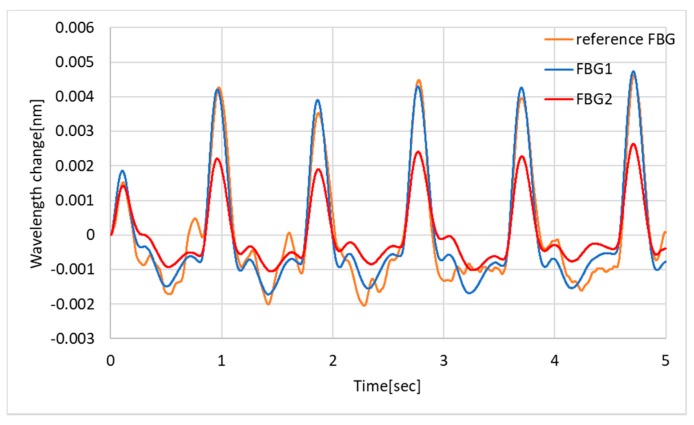
A Butterworth-type bandpass filter (order 2, low-cutoff frequency 0.5 Hz, high-cutoff frequency 5 Hz) is applied onto the data of Figure 9. The offset value of the reference FBG is subtracted before filtering.

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
