# Peer review of "Wireless, Portable Fiber Bragg Grating Interrogation System Employing Optical Edge Filter"

_sensors, 2019, doi:10.3390/s19143222_

Round 1
Reviewer 1 Report
The authors presented a portable device for interrogation of fiber optic Bragg grating. The WDM Edge filter is used for conversion from wavelength shift to light intensity. The article is written correctly and in accordance with the scientific methodology. In addition to the hardware solution (main achievement ?), it also contains measurements and its analysis.
1. What is the main scientific accomplishment of this work?
2. Is the presented differential system insensitive to fluctuations in the level of the radiation source?
3. Why use the first and second derivative in the plethysmograph data analysis?
4. What are the parameters of the Butterwoth filter used to reduce the signal noise?
5. What spectrum does BBS light source have?
6. It would be worth to slightly extend the analysis of the measurements and received signals.
Author Response
Thank you for your useful comments.
Point 1: What is the main scientific accomplishment of this work?
Response 1: The combination of edge filter and newly designed amplifier circuit aimed to measure the plethysmograph with FBG(s). I added a sentence on the abstract to clarify the point.
Point 2: Is the presented differential system insensitive to fluctuations in the level of the radiation source?
Response 2: It is affected, however some reasons it doesn't affect on the measurement.
(1) Generally the fluctuation is slower than the pulse wave (plethysmogram).
(2) The differential reduce the influence of the fluctuation.
(3) Only the dynamic signal is amplified and used finally.
Point 3: Why use the first and second derivative in the plethysmograph data analysis?
Response 3: Ordinary these analysis is used to obtain the feature of the waveform. However, I felt it is unnecessary in this situation so I erased the graphs.
Point 4: What are the parameters of the Butterwoth filter used to reduce the signal noise?
Response 4: I added the parameters on the manuscript.
Point 5: What spectrum does BBS light source have?
Response 5: I added it on Figure 1.
Point 6: It would be worth to slightly extend the analysis of the measurements and received signals.
Response 6: Instead of the derivative, I added a reference data obtained with commercially available interrogator. It shows the higher signal to noise ratio, and repeatability.
Reviewer 2 Report
The authors presented an FBG interrogator based on edge-filter technique for static and dynamic measurements. To validate the system, the plethysmograph signal of the radial artery is obtained with an FBG.
In my opinion, despite a good set of experiments and analysis are presented, the authors omit important information related with the interrogator. In addition, the manuscript has a lot of typos. The authors are advised to resubmit the manuscript in Sensors MDPI Journal after the following suggestions are addressed:
Please, separate the values from the unit. For instance, 0.2 pm instead 0.2pm
Page 2, line 53. (3) passive instead (3)passive; line 56 multiplexing (WDM) instead multiplexing(WDM); line 57 Mach-Zehnder [8,35] instead Mach-Zehnder[8,35]. Please carefully edit your manuscript by correcting typos, and please have a person fluent in English proofread your paper.
Pages 3 and 4, lines 86 and 112, respectively. WDM was already defined in Page 2, line 56.
Figure 3 and Figure 5 are exactly the same. There is on new information in Figure 5. The caption figure is referring to (a) and (b). Please, verify and include the correct figure.
Page 4, line 110. Could the authors include more details about which type of SLD, PDs, WDM, and half mirror were used. For instance, (reference, brand)
It is not clear if the WDM is used as an edge-filter, or are used both the WDM and the edge-filters, please explain better.
Could the authors present the spectra of both edge-filters and FBGs? Are the FBGs and edge-filters spectra centered at the same wavelength (1543 nm and 1561 nm)? Verify that FBGs are affected by temperature effects.
Page 4, line 131, Equation 2 instead (2); line 139, FBGs instead FBG.
Could the authors give more details about interrogator, such as size, weight, power consumption
The authors claim the interrogator can measure simultaneously two FBGs. What about the results of second FBG?
Regarding Figure 6. Please, explain better, define OSA and put names on the picture.
Regarding Figure 7. If both FBGs are submitted to axial strain, why the second FBG is negative (lower graph)?
What about the temperature cross-sensitivity? Please, discus about temperature effects.
Author Response
Thank you for your detailed comments.
Point 1:
Please, separate the values from the unit. For instance, 0.2 pm instead 0.2pm
Page 2, line 53. (3) passive instead (3)passive; line 56 multiplexing (WDM) instead multiplexing(WDM); line 57 Mach-Zehnder [8,35] instead Mach-Zehnder[8,35]. Please carefully edit your manuscript by correcting typos, and please have a person fluent in English proofread your paper.
Pages 3 and 4, lines 86 and 112, respectively. WDM was already defined in Page 2, line 56.
Page 4, line 131, Equation 2 instead (2); line 139, FBGs instead FBG.
Regarding Figure 6. Please, explain better, define OSA and put names on the picture.
It is not clear if the WDM is used as an edge-filter, or are used both the WDM and the edge-filters, please explain better.
Response 1: I have revised these points. Thank you for your indication.
Point 2: Figure 3 and Figure 5 are exactly the same. There is on new information in Figure 5. The caption figure is referring to (a) and (b). Please, verify and include the correct figure.
Response 2: I apologize about insufficient confirmation. I changed it to the figure I originally intended.
Point 3: Page 4, line 110. Could the authors include more details about which type of SLD, PDs, WDM, and half mirror were used. For instance, (reference, brand)
Response 3: I added the detail of the PDs and the half mirror, and the spectrum of the SLD on Figure 1. The WDM filter is only used as dividing the wavelength range of FBGs, so I think it makes complicated if the spectrum is added on the manuscript. Please see the attached file.
All these parts except PDs are custom-made, impossible to show the brand name or reference, so I inscribed characteristics as detailed as possible in the revised manuscript.
Point 4(1): Could the authors present the spectra of both edge-filters and FBGs? Are the FBGs and edge-filters spectra centered at the same wavelength (1543 nm and 1561 nm)? Verify that FBGs are affected by temperature effects.
Point 4(2): What about the temperature cross-sensitivity? Please, discus about temperature effects.
Response 4: I added the spectrum of edge-filters and FBGs. Of course the FBGs are affected by temperature as 9.8 pm/C, however in the application of the plethysmograph cause only about 10 C temperature shift. The edge filter has about 1 nm range to be used (Figure 7) so it is not a problem.
Point 5: Could the authors give more details about interrogator, such as size, weight, power consumption
The authors claim the interrogator can measure simultaneously two FBGs. What about the results of second FBG?
Response 5: I added the weight and power consumption. The size is written on the original manuscript.
I also added the result of two FBGs. I used another individual of the interrogator so I updated the coefficients of the formula.
Point 6: Regarding Figure 7. If both FBGs are submitted to axial strain, why the second FBG is negative (lower graph)?
Response 6: This is because the slope of reflection/transmission ratio around the used wavelength have opposite sign. It is shown in Figure 7 (a) and (b).

Reviewer 3 Report
The authors propose and experimentally demonstrate a compact Fiber Bragg Grating Interrogation System employing optical edge filter, showing the plethysmograph monitoring. The system concept and the measured data are analysed and compared. The paper would be of interest for readers in the field of both plethysmograph monitoring and FBG sensing application. There are some following matters to be properly addressed.
1 The FBG reflection spectrum could be presented. It is even better to include the real-time shift of the spectrum during the measurement.
2 What are the physical indications of α and β in Eq.(2)?
3 Figure 5 is duplicated and incorrect.
4 Since the authors claim the measurement of slower-changing signals such as temperature, the data for temperature monitoring should be presented.
5 What do a/d/c/d/e points mean in Fig.11(b)?
6 The manuscript write-up and format is to be improved. It is necessary to carefully edit the manuscript by correcting typos.
Author Response
Thank you for your useful comments.
Point 1: The FBG reflection spectrum could be presented. It is even better to include the real-time shift of the spectrum during the measurement.
Response 1: I added these graphs in Figure 1 and Figure 7.
Point 2: What are the physical indications of α and β in Eq.(2)?
Response 2: I added the explanation in the manuscript. These are the coefficient determined by the characteristics of edge filter.
Point 3: Figure 5 is duplicated and incorrect.
Response 3: I apologize about my insufficient confirmation. I changed it to the figure I originally intended.
Point 4: Since the authors claim the measurement of slower-changing signals such as temperature, the data for temperature monitoring should be presented.
Response 4: I erased the description about temperature from the abstract because it is not the purpose of this study and confusing. In this study I only used function to obtain the value of alpha and beta, and the temperature measurement is only the possible application. Instead I added the description about the circuit which designed to amplify effectively the plethysmograph signal which is the main novelty of this study.
Point 5: What do a/d/c/d/e points mean in Fig.11(b)?
Response 5: These points which indicate the peaks/valleys of second derivation of the pulse wave signal is used to extract the feature of the finger plethysmogram waveform. However, I felt it is unnecessary in this situation so I erased the graphs. Instead, I added the comparison against commercially available interrogator.
Point 6: The manuscript write-up and format is to be improved. It is necessary to carefully edit the manuscript by correcting typos.
Response 6: Thank you for your indication and I checked it again.
Round 2
Reviewer 2 Report
The authors have addressed correctly my suggestions and the paper have been enhanced considerately. In my opinion, the manuscript can be accepted for publication in Sensors MDPI Journal after the following suggestions are addressed:
Regarding Figure 1. Please, use (a), (b) and (c) indicators to refer each figure. In addition, introduce them in the text. Since the spectra are real instead schematic representation, it is important to describe which device was used to measure them. For instance, an OSA or commercial interrogator. Please, verify that measure device resolution modifies the spectra shape. Therefore, include the resolution value.
Could the authors mesh the Figures 2 and 3? Please, explain with more details the Figure 3. Despite the interrogation concept is simple, whether a figure is exposed, it is expected its explanation.
Could the authors explain some limitations of the proposed approach?
Regarding Figures 4 and 5. Could the authors explain what is the function of PD5?
The authors are advised to review next references
C. A. Díaz, C. A. Marques, M. F. F. Domingues, M. R. Ribeiro, A. Frizera-Neto, M. J. Pontes, P. S. André, and P. F. Antunes, "A Cost-Effective Edge-Filter Based FBG Interrogator Using Catastrophic Fuse Effect Micro-cavity Interferometers", Measurement, vol. 124, pp. 486-493, Apr. 2018.
C. A. R. Díaz, C. Leitão, C. A. Marques, M. F. Domingues, N. Alberto, M. J. Pontes, A. Frizera, M. R. N. Ribeiro, P. S. B. André, and P. F. C. Antunes, "Low-cost interrogation technique for dynamic measurements with FBG-based devices", Sensors, vol. 17, no. 2414, Oct. 2017.
Author Response
Thank you for your comments.
Point 1: Regarding Figure 1. Please, use (a), (b) and (c) indicators to refer each figure. In addition, introduce them in the text. Since the spectra are real instead schematic representation, it is important to describe which device was used to measure them. For instance, an OSA or commercial interrogator. Please, verify that measure device resolution modifies the spectra shape. Therefore, include the resolution value.
Response 1: I added these indicators and the resolution of OSA.
Point 2: Could the authors mesh the Figures 2 and 3? Please, explain with more details the Figure 3. Despite the interrogation concept is simple, whether a figure is exposed, it is expected its explanation.
Response 2: I combined Figure 2 and Figure 3 as Figure 2(a) and (b), and added the explanation of the slopes.
Point 3: Could the authors explain some limitations of the proposed approach?
Response 3: I added the sentence below in the introduction.
" The limitation of this type is the range of Bragg wavelength, however this disadvantage is not a problem for plethymogram, because the signal amplitude is totally within the range. "
Point 4: Regarding Figures 4 and 5. Could the authors explain what is the function of PD5?
Response 4: The PD5 on Figure 3 (former Figure 4) (not depicted in Figure 4 (former Figure 5)) is only used for auto power control of the SLD, and it doesn't noticeably affect on the measurement result here.
Point 5: The authors are advised to review next references
C. A. Díaz, C. A. Marques, M. F. F. Domingues, M. R. Ribeiro, A. Frizera-Neto, M. J. Pontes, P. S. André, and P. F. Antunes, "A Cost-Effective Edge-Filter Based FBG Interrogator Using Catastrophic Fuse Effect Micro-cavity Interferometers", Measurement, vol. 124, pp. 486-493, Apr. 2018.
C. A. R. Díaz, C. Leitão, C. A. Marques, M. F. Domingues, N. Alberto, M. J. Pontes, A. Frizera, M. R. N. Ribeiro, P. S. B. André, and P. F. C. Antunes, "Low-cost interrogation technique for dynamic measurements with FBG-based devices", Sensors, vol. 17, no. 2414, Oct. 2017.
Response 5: Thank you for your recommendation. I added these articles in the sentence about the prior research of edge-filter based FBG measurement.
Response 6: Also, I found mistakes on my manuscript so I revised it too.
(1) In section 3.1, I mistakenly wrote the size of commercially available FBG interrogator. So I changed it to the size of developed interrogator.
(2) I didn't write about the wavelength measurement of Figure 6(former Figure 7), so I added a sentence about OSA.
Reviewer 3 Report
Accepted
Author Response
Thank you for accepting my manuscript.
I revised several points the other reviewer pointed.
Also, I found mistakes on my manuscript so I revised it too.
(1) In section 3.1, I mistakenly wrote the size of commercially available FBG interrogator. So I changed it to the size of developed interrogator.
(2) I didn't write about the wavelength measurement of Figure 6(former Figure 7), so I added a sentence about OSA.